# Gold Nanoparticles as Contrast Agents in Ophthalmic Imaging

Alexandra Kavalaraki [1], Ellas Spyratou [1], Maria Anthi Kouri [1,2] and Efstathios P. Efstathopoulos [1,*]

1.   2nd Department of Radiology, Medical School, National and Kapodistrian University of Athens, 11527 Athens, Greece
2.   Medical Physics Program, University of Massachusetts Lowell, 265 Riverside St, Lowell, MA 01854, USA
*   Correspondence: stathise@med.uoa.gr; Tel.: +30-210-583-1818

**Abstract:** Over the past few years, tremendous research concerning the possibilities of gold nanoparticles in medicine has been conducted. Gold nanoparticles (AuNPs) are considered to be unique nanostructures due to their extraordinary chemical and physical properties. This review article aims to bring into light the potential applications of gold nanoparticles for diagnostic purposes in ophthalmology. More specifically, attention will be drawn to the utilization of AuNPs as contrast agents (CAs) in optical coherence tomography (OCT) and photoacoustic imaging (PAI), which are two novel imaging modalities for the visualization of the eye. None of these techniques requires the use of an imaging adjuvant to function; however, the addition of a contrast agent has been proposed for image improvement, and AuNPs are attractive candidates for this purpose. The in vitro, ex vivo, and in vivo studies investigating and supporting this concept will be presented thoroughly to elucidate whether AuNPs are eligible for imaging enhancement owing to their optical characteristics.

**Keywords:** contrast agents; ophthalmic imaging; ophthalmology; optical coherence tomography; photoacoustic imaging; nanotechnology; gold nanoparticles

## 1. Introduction

Nanotechnology is one of the fastest advancing fields in scientific research. In 1959, Richard P. Feynman stated during the annual American Physical Society meeting that "There's Plenty of Room at the Bottom", in an attempt to present the enormous possibilities offered within this field [1]. Among the medicinal branches exploiting the advantages of nanomaterials and nanotechnology, ophthalmology can also benefit from the introduction of this emerging science. According to the World Health Organization, blindness and visual impairment affect more than 2.2 billion people worldwide, posing a paramount social and economic burden. Nanotechnology can accelerate the progress toward imaging and therapy in ophthalmology, and nanoparticles have already been employed for ocular drug delivery due to their ability to be engineered in a way that allows them to transcend the physical and anatomical barriers of the eye [2,3].

This review focuses on the potential applications of gold nanoparticles in ophthalmic imaging modalities. Gold nanoparticles (AuNPs) exhibit optical properties that can be advantageous for ocular imaging with optical coherence tomography (OCT) and photoacoustic imaging (PAI). OCT is a non-invasive imaging modality that detects the backscattering of light from biological tissues and allows the cross-sectional visualization of the retinal layers [4]. PAI of biological tissues is a hybrid technology based on the detection of acoustic waves generated from the absorption of optical energy and appears to be promising for ophthalmic imaging [5]. The introduction of a contrast agent such as AuNPs in the above imaging modalities can promote molecular imaging in ophthalmology and allow the diagnosis of ocular diseases at an early stage, before vision loss takes place. Gold nanoparticles also exhibit anti-angiogenic properties and could be useful for the management of ocular diseases, characterized by neovascularization [6]. Therefore, gold nanoparticles are attractive candidates to be used as "theranostics" in ophthalmology, which describes their ability

to improve both ophthalmic imaging as well as the management of neovascular conditions in the eye.

## 2. Results

### 2.1. Imaging in Ophthalmology

The eye is a complex and highly specialized sensory organ, whose main functions involve the detection and focus of visual stimuli, as well as the conversion of light into electrical signals that are conveyed to the brain, in order to form an image and initiate the visual process [7]. Over the last two decades, ophthalmic imaging has been gaining significant importance, becoming an indispensable part of the clinical diagnosis and management of various ocular diseases, affecting both the anterior and posterior segment of the eye [8]. Two novel imaging modalities for ocular visualization are optical coherence tomography (OCT) and photoacoustic imaging (PAI).

#### 2.1.1. Optical Coherence Tomography

Optical coherence tomography (OCT) was first introduced in 1991 by Huang et al. [9]. They presented a non-invasive technique that uses low-coherence interferometry and achieves the cross-sectional 2D imaging of internal biological structures by measuring their optical reflections. The system is based on a Michelson interferometer, which calculates the interference signal from the sample and the reference beam. Optical signals transmitted through or reflected from tissues provide information acquired from the time-of-flight delay, which in turn offers spatial data of the tissue microstructure. Most OCT systems use near-infrared (NIR) light (~850 nm), since it has the benefit of deeper tissue penetration [10,11]. OCT offers a high depth resolution, which is particularly useful for imaging deep tissues of the eye, such as the intraretinal layers [12–14]. This imaging modality has become a key diagnostic tool in the field of ophthalmology and is considered to be the gold standard for the detection of a wide range of diseases and pathological conditions of both the anterior and posterior segment of the eye.

#### 2.1.2. Photoacoustic Imaging

Photoacoustic imaging (PAI), also known as optoacoustic imaging, is an emerging non-ionizing imaging modality, which is characterized by the combination of the high contrast of optical imaging as well as the high spatial resolution of ultrasound (US) [15,16]. It is a non-invasive technology based on the photoacoustic effect, which was first observed by Alexander G. Bell in 1880 [17]. The PA effect concerns the acoustic wave production as a result of the optical irradiation of biological tissue through a pulse laser beam. Endogenous chromophores or exogenous contrast media absorb the energy of the light, and heat is released due to thermoelastic relaxation. The occurring pressure waves, named PA waves, are then detected by the US transducer, and the image is reconstructed. PAI is a technology that merges light and sound, and the term "Light In and Sound Out" is therefore used to describe this technique [16].

Photoacoustic imaging differs from other imaging modalities such as OCT and US, in the sense that it portrays optical absorption and is not influenced by the mechanical and elastic characteristics of the tested tissue, and it allows us to gain anatomical, structural as well as molecular knowledge of the tissue [18–20]. The application of PAI in different medical fields has been encouraged, and its introduction in ophthalmology has also been suggested and appears to be promising, due to the presence of endogenous light-absorbing molecules within the eye: melanin and hemoglobin [21,22]. PAI exhibits various benefits; however, it has not yet been implemented in ophthalmology. Research groups endeavor to design an ocular photoacoustic imaging system that could be applicable to humans, but further investigation is necessary.

### 2.1.3. Molecular Ophthalmic Imaging

Even though a variety of ocular imaging modalities are currently being used in clinical practice, there are still limitations to be dealt with. Morphological changes linked to diseases are often revealed and can be detected after functional and molecular changes have already manifested [23]. This is particularly seen in retinal diseases, leading to vision loss before alterations in the tissue can be visualized [24]. Therefore, there is the need for the development of molecular imaging, which aims to detect functional as well as molecular changes in the eye. For this purpose, the introduction of exogenous contrast agents in traditional imaging modalities has been suggested, since they can enhance the signal and help us visualize and quantify molecular and biological processes non-invasively [25,26].

OCT and PAI are imaging techniques that do not require the use of an exogenous contrast agent; however, the concept of adding an exogenous imaging agent to improve image quality and enhance contrast has been a subject of great interest for researchers. Over the years, several imaging agents have been suggested, including microbubbles [27] and microspheres [28], indocyanine green [29], near-infrared dyes [30,31] and various types of NPs [32–34]. Gold nanoparticles (AuNPs) are one of the most studied types of nanostructures for application in bioimaging, due to their highly favorable properties for in vivo imaging, and they have also been proposed as contrast agents for both OCT and PAI [35].

### 2.2. Gold Nanoparticles

#### 2.2.1. Preparation Methods of Gold Nanoparticles

During the 19th century, Michael Faraday was the first scientist to report the synthesis of colloidal gold nanomaterials in the literature [36]. He noticed that the use of phosphorus as a reduction agent for gold chloride produced particles that created a "beautiful ruby fluid", resulting from their interaction with light. This inspired the theoretical work by a German physicist called Gustav Mie, who explained that the ruby color of a spherical AuNP solution relates to the absorption and scattering of the light interacting with it [37].

Regarding the synthesis of nanomaterials nowadays, two different techniques can be applied: either a "bottom-up" or a "top-down" strategy [38]. Spherical gold nanoparticles are principally synthesized with a "bottom-up" technique, using a reducing agent such as $NaBH_4$ or sodium citrate, for the reduction of Au (III) ions [39]. The concept of citrate reduction of Au (III) to Au (0) in water was first established by Turkevitch et al. [40] during the 20th century, and this technique is still employed nowadays.

Gold nanoparticles can be stabilized with the addition of various molecules through thiol–gold interactions [41]. Giersig and Mulvaney [42] revealed that AuNPs can be stabilized by thiolates via a sulfur bond. The Shiffrin–Brust biphasic synthesis, which is commonly used, is based on sulfur coordination and uses $HAuCl_4$, a thiol, tetraoctylammonium bromide and $NaBH_4$ [43]. To gain better stability, surfactants such as CTAB (cetyltrimethylammonium bromide) or agents that control the surface are often added [44]. Such surfactants are also used for the fabrication of anisotropic AuNPs [45]. Apart from the selection of the appropriate surfactant, the use of shape-regulating molecules is also employed for the preparation of AuNPs with different geometries [46]. The selection of the solvent and its interaction with the nanoparticle surface is also a determinant for the NP morphology [47]. Even though numerous preparation methods of different AuNPs are applied, the seed-mediated method is the most commonly used, especially for the fabrication of gold nanorods [45].

#### 2.2.2. Optical Properties of Gold Nanoparticles

A unique optical feature of AuNPs results from the collective oscillation of the metal's conductive electrons when an electromagnetic field is applied. The oscillation of the free electrons is driven by an alternating electric field at an eigenfrequency (ωp), which is relative to the lattice of positive ions [48]. This phenomenon is commonly known as localized surface plasmon resonance (LSPR), as it leads to the formation of a "negatively charged

cloud", the localized surface plasmon. This process involves two types of interactions between light and matter: scattering, which results in the light being re-radiated in different directions, but at the same frequency, and absorption, which causes the conversion of light to heat [39]. The way Au nanoparticles interact with light also explains the intense colors that they present.

The LSPR parameters of a gold nanostructure (peak position and scattering-to-absorption ratio) are highly dependent on a variety of parameters including size, shape, aggregation, morphology, and surrounding environment [49,50]. The alteration of these factors enables the shift of the absorption band within the visible and to the NIR, which is beneficial for optical imaging [51]. LSPR shifts to the NIR can easily be achieved by utilizing anisotropic/non-symmetrical AuNPs, which explains the great interest in them for biomedical applications. Among anisotropic NPs, gold nanorods are probably the most studied type due to their optical tunability by altering their aspect ratio [52] (Figure 1). Due to their morphology, the conductive electrons of gold nanorods are able to oscillate in two different directions resulting in a transverse and longitudinal LSPR peak, with the latter tuning into the NIR [53]. As a result, the aspect ratio (the ratio of the length of the long axis to the short axis) is a determinant parameter for the adjustment of the two separate bands that control the optical properties of gold nanorods. The longitudinal band is shifted from the visible to the NIR when the aspect ratio is raised, whereas the transverse band undergoes small blue-shift variations [54]. Other anisotropic AuNPs that present highly plasmonic properties include nanoshells [55], nanocages [56], nanostars [57], and nanodisks [58]. The exceptional physical properties of AuNPs, in combination with their facile fabrication and functionalization, render them appropriate for in vivo bioimaging applications [59].

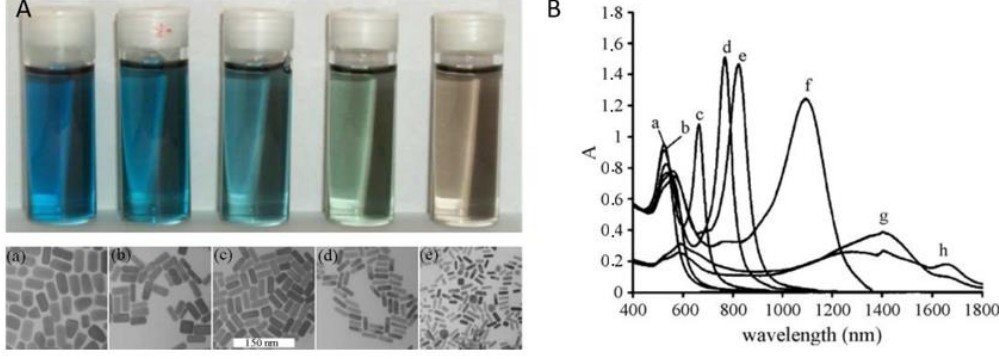

**Figure 1.** (**A**) The changes in aspect ratio of AuNRs induce drastic changes in transmitted colors as the aspect ratio increases (**a**–**e**). Reprinted from publication [52] Coord. Chem. Rev, vol. 249, 17–18 SPEC. ISS., J. Pérez-Juste, I. Pastoriza-Santos, L. M. Liz-Marzán, and P. Mulvaney, "Gold nanorods: Synthesis, characterization and applications," pp. 1870–1901, Copyright (2005), with permission from Elsevier (**B**) UV–vis spectra of Au nanorods with increasing aspect ratios (**a**–**h**). As the aspect ratio of AuNRs increases, the absorption spectrum shifts to near-infrared wavelengths. Reprinted from publication [60] Adv. Mater. Vol 13, N. R. Jana, L. Gearheart, C. J. Murphy, Seed J. Murph Growth Approach for Shapeth Approach Synthesis of Spheroidal and Rodnthes Gold Nanoparticles Using a Surfactant Template" pp. 1389–1393, copyright (2001) WILEY1389-Verlag GmbH, Weinheim, Fed. Rep. of Germany.

The ability of AuNPs to customize their optical properties and extinction band has brought attention to their use in various imaging techniques, including X-ray computed tomography, dark field microscopic imaging, magnetic resonance imaging, and fluorescence imaging [61–63]. Aside from the above modalities, AuNPs are also considered to be attractive candidates for imaging enhancement using techniques such as OCT and PAI [64–66].

As explained previously, OCT images result from the detection of backscattering of light from biological tissues. Gold nanoparticles exhibit excellent light scattering ability, which is many times stronger than that generated by conventional fluorophores. Therefore,

imaging enhancement can be achieved by tuning the optical properties of AuNPs in the wavelength, in which the OCT system operates. Furthermore, gold nanostructures can effectively act as imaging adjuvants in photoacoustic imaging. Cross-sectional photoacoustic images of biological tissues are based on the principles of optical and ultrasonic/acoustic waves, which means that acoustic waves are generated after the absorption of optical energy. The operation of this system, therefore, requires the presence of light-absorbing molecules. Gold nanoparticles can effectively absorb light in a tunable manner and can consequently serve as exogenous contrast agents for this imaging modality [23].

The facile adjustment of AuNPs' optical properties allows them to either cause high backscattering of light or absorb light effectively, which is beneficial for imaging with OCT and PAI, respectively. Figure 2 is a schematic representation of the application of AuNPs as contrast agents for these novel imaging techniques, as illustrated by Chen et al. [23].

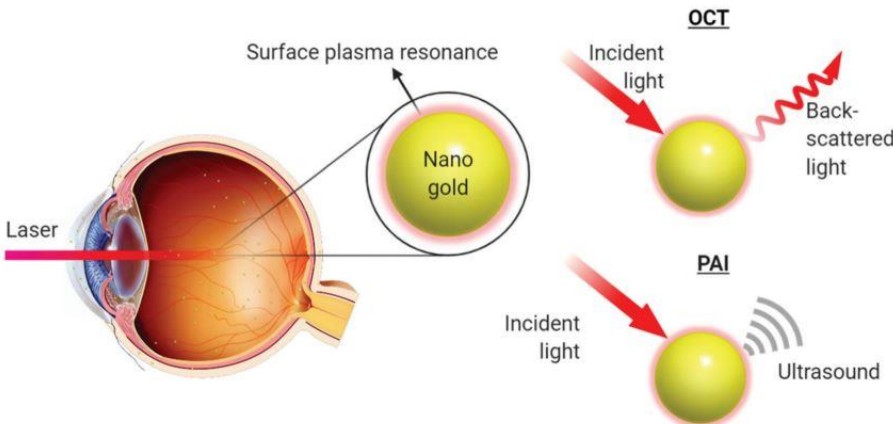

**Figure 2.** Schematic illustration of the use of gold nanoparticles as contrast agents in optical coherence tomography and photoacoustic imaging for the visualization of the eye. After the stimulation with incident light, gold nanoparticles can create backscattered light for detection by the OCT camera or an ultrasound signal to be detected by the PAI system. Plasmons convert a part of the oscillation energy into heat, which is then detected by the PAI system. Copyright © 2021 Biomater Sci. Author manuscript; available in PMC 21 Jan 2022. Published in final edited form as: Biomater Sci. 21 Jan 2021; 9(2): 367–390. Published online 15 Oct 2020. doi:10.1039/d0bm01063d [23].

*2.3. Studies Investigating the Use of Gold Nanoparticles as Contrast Agents (CAs) for OCT and PAI*
2.3.1. Gold Nanoparticles as Contrast Agents for Optical Coherence Tomography
In Vitro Studies

Gold nanorods (AuNRs/GNRs) are generally considered to be beneficial as contrast agents for in vivo imaging, because they present an intense and narrow LSPR absorption band in the NIR, which can be easily tuned depending on their aspect ratio [67]. The application of cylinder-like-shaped Au nanorods (diameter: 20 nm, LSPR peak: 750 nm, 912 nm) as backscattering CAs in a time-domain OCT was tested by Troutman et al. [68] using tissue phantoms. The LSPR peak was close to the OCT spectral distribution, leading to a much stronger signal. They, furthermore, noticed that both of the nanorod suspensions examined enhanced the signal contrast compared to water, indicating that AuNRs effectively worked as contrast agents for this OCT system.

The effectiveness of AuNRs (diameter: 10 nm) as CAs for a Fourier-domain OCT system was determined by Jia et al. [69], using tissue phantoms and cultured retinal pigment epithelium (RPE) cells. OCT images from the intralipid alone were compared with those obtained by the AuNR sample and the intralipid mixed with GNRs, and it was proved that the presence of AuNRs caused a red shift. Then, images of 1% gelatin, unlabeled RPE cells, and RPE cells labeled with PEG- and Tat-coated GNR (10 × 50 nm) were taken. The spectral shift allowed the visualization of the labeled RPE cells. The

purpose of this study was to design a cellular CA, with the purpose of finding a method to track stem cells and promote the management of retinal diseases in the future.

AuNRs have also been studied as contrast agents for OCT imaging with an operation wavelength around 1300 nm by Ratheesh et al. [70] using tissue phantoms. These AuNRs presented a high aspect ratio of 8.8 and their LSPR peak was estimated at 1320 nm. It was proved that these nanostructures can effectively act as absorption-based imaging adjuvants in this wavelength, due to their high absorption/scattering ratio.

The use of AuNRs in a dual-band OCT system with two separated bandwidths was discussed by Rawashdeh et al. [71]. GNRs of different dimensions were studied using a highly scattering agar phantom. Their idea was to measure the different contrast produced when the NPs used were only resonant to one wavelength of the dual system. The strongest differential contrast at both wavelengths was produced with the aid of a sample of large nanorods (length: 75 nm), whereas signal intensities were not detectable with smaller dimensions (length: 48 nm).

The use of large gold nanorods (LGNRs) as spectral OCT contrast enhancers was also proposed by Liba et al. [72,73]. Their larger dimensions (~100 × 30 nm) offer advantages compared to smaller nanorods, due to the ability to produce stronger backscattered and spectral signals. Through their in vivo study, they discovered that these LGNRs presented a 30-fold higher OCT intensity and were able to produce a spectral signal per particle more than 100 times stronger compared to conventional GNRs.

Gold nanoshells have also been proposed as OCT contrast agents due to their high scattering and low absorption efficiency [74]. Agrawal et al. [75] performed quantitative measurements to investigate OCT contrast enhancement using Au nanoshells. Imaging was performed in water and tissue-simulating phantoms. Mono-layered nanoshells of different concentrations and geometries were evaluated, and the results revealed a monotonic elevation of the OCT signal intensity and attenuation when the shell and core size were enlarged. The nanoshells leading to the strongest backscattering were found to have a core diameter of 291 nm and shell thickness of 25 nm, and the threshold concentration for signal elevation (2 dB) was measured at $10^9$ nanoshells/mL. They, therefore, proved that Au nanoshells are promising CAs for OCT image improvement by optimizing their dimensions.

An OCT system was utilized by Zagaynova et al. [76] to evaluate the contrasting abilities of silica–gold nanoshells as contrast media. The nanoshells exhibited a silica core size of 150 nm and gold shell thickness of 25 nm, and agar biotissue phantoms were chosen for the experiment. OCT imaging proved that gold nanoshells penetrating the phantoms caused the intensification of the signal.

Gold nanoshells (core: 120 nm, shell thickness: 16 nm) were also tested as an exogenous CA for a phase-sensitive OCT imaging system in vitro by Adler et al. [77]. Photothermal modulation was induced to study the modifications of the optical path length caused by temperature oscillations and OCT phase microscopy was used on pure deionized water and an Au nanoshell solution. The results illustrated a high contrast between the phantoms with and without NPs. Consequently, nanoshells are also prospective candidates as contrast agents in a phase-sensitive OCT system.

Gold nanocages are nanoparticles with a hollow and porous morphology. They are usually fabricated through the galvanic replacement reaction between silver nanocubes and $HAuCl_4$ in solution and have also been suggested as CAs [42]. Gold nanocages (average length: ~35 nm) were propose as potential contrast media for spectroscopic OCT in vitro, using gelatin-made tissue phantoms, by Cang and co-workers [78].

Gold nanoparticles with a star-like morphology, also referred to as gold nanostars (GNSs), have also been engineered to be studied as contrast agents. Ponce-de-Leon et al. [79] fabricated AuNPs of spherical-, cubic- or star-like-shaped geometry and analyzed their ability to enhance OCT imaging. Among these, the AuNPs with a morphology resembling a star and having sizes less than 150 nm produced the best contrast in water as well as in

agarose phantoms. This was the first time that gold nanostars were introduced as potential contrast materials for OCT contrast enhancement.

The use of Au nanostars as contrast adjuvants for OCT and Doppler OCT was evaluated by Bibokova et al. [80]. GNSs of various sizes and different numbers of spherical seeds were used for this study, and it was remarked that the most promising contrast enhancers were the large-sized nanostars (120 nm), due to their prominent scattering properties. Therefore, by altering the number of seeds and consequently the size of the nanostars, it was possible to adjust their scattering properties to the appropriate wavelength, which renders them appealing as imaging agents.

Ex Vivo Studies

Wang and co-workers [81] performed AuNR-enhanced Doppler OCT scans to image the intrascleral aqueous humor outflow. The aqueous flow, which normally does not produce a Doppler signal, determines intraocular pressure and is therefore related to the development of glaucoma [82]. A solution of gold nanorods (~40 × 10) at a concentration of $1 \times 1012$ AuNRs/mL was injected into porcine eyes with mock aqueous, whereas the control group received injection of Barany's mock aqueous. The presence of Au nanorods produced a measurable signal, allowing the visualization of the anterior chamber outflow.

In a study by Prabhulkar et al. [83], antibody-conjugated Au nanorods were examined as backscattering OCT contrast adjuvants for molecular histopathology on an ocular surface squamous neoplasia (OSSN) model. Anti–glucose transporter-1 (Glut-1), which is overexpressed in OSSN lesions, was selected as a molecular target for the NP functionalization [84]. The term OSSN describes a variety of pathological entities and lesions, including dysplasia of the cornea and conjunctiva epithelium, carcinoma in situ (CIS), and invasive squamous-cell carcinoma (SCC) [85].

OCT imaging of the control specimens (no epithelial atypia) revealed a minimal background signal in the epithelium. OCT images of the conjunctival specimens with CIS disclosed a weak background signal within the epithelium, even though immunofluorescence showed intense staining. In the three cases of SCC studied, the OCT of only two showed increased signal agreeing with the immunofluorescence-positive regions. No imaging enhancement was seen in the third case, even though there was moderate staining in immunofluorescence. The above research findings suggest the presence of a threshold concentration of AuNRs for the generation of a detectable OCT signal. Despite the limitations of this study, it is the first reported application of Au-enhanced imaging for this purpose and broadens the scope of research for the use of molecular markers in ocular imaging.

Faber et al. [86] discussed the use of nanoshells (core radius: 290 nm; gold shell thickness: 33 nm) as contrast agents for OCT in an attempt to develop nanoparticle-assisted optical molecular imaging (NAOMI). The NPs were injected subretinally into porcine eyes, and the withdrawal of the syringe created a "cloud" of nanoshells, which appeared to sediment onto the retina and create a reflective layer. These observations supported the idea of using Au nanoshells as an OCT contrast agent.

Ozone ($O_3$) has oxidizing effects and can be harmful when penetrating the damaged corneal epithelium of the eye [87]. To visualize the penetration of $O_3$ into the anterior chamber of the eye, Jiang et al. [88] used Au triangular nanoprisms as CAs for 3D OCT imaging, using an isolated crucian carp eye. The idea to use AuTNPs as CAs arose from the blue shift of the plasmon peak due to morphological changes in the presence of $O_3$ [89]. After the Au nanoparticle injection (~0.0228 mg/mL) in the anterior chamber of the eye, contrast enhancement was noticed in OCT imaging. The crucian carp eye with a damaged cornea was exposed to $O_3$ to study its distribution and OCT images were obtained at several time points. It was noted that the area around the wound had enhanced contrast due to the NPs' morphological change (Figure 3). These results indicated that Au nanoprisms could serve as a promising CA for the detection of $O_3$ in the eye using OCT imaging.

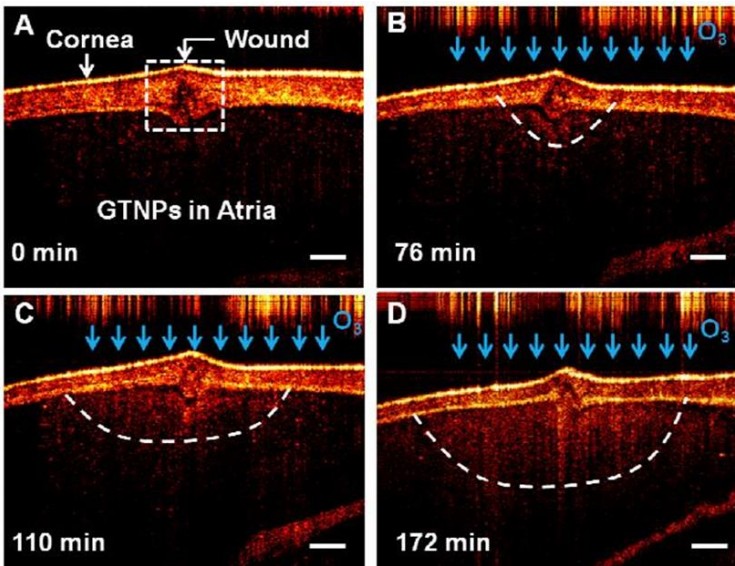

**Figure 3.** Images of an isolated crucian carp eye after the AuTNP injection under ozone exposure: 0 min (**A**), 76 min (**B**), 110 min (**C**), and 172 min (**D**). The scale bars represent 300 μm. Reprinted with permission from X. Jiang, P. Tang, P. Gao, Y. S. Zhang, C. Yi, and J. Zhou, "Gold Nanoprobe-Enabled Three-Dimensional Ozone Imaging by Optical Coherence Tomography," Anal. Chem., vol. 89, no. 4, pp. 2561–2568, 2017, doi:10.1021/acs.analchem.6b04785. Copyright © 2017, American Chemical Society [88].

In Vivo Studies

De la Zerda et al. [90] investigated the use of gold nanorods (GNRs) as optical coherence contrast agents in vivo, in mice eyes. Corneal and anterior chamber injections of two different sizes of GNRs were performed: GNRs corresponding to a peak absorbance wavelength of 780 nm (GNR-780) and 850 (GNR-850). When GNR-850 was injected into the anterior chamber at various concentrations, it was seen that concentrations of 120 pM or above resulted in a significantly different OCT contrast (Figure 4). Moreover, the signal from corneas injected with GNR-780 was three times stronger compared to mice corneas injected with balanced saline solution (BSS) and 7.5 times stronger compared to naïve mice corneas (Figure 5). Overall, this study led to the conclusion that GNRs can be characterized as high-sensitivity CAs for OCT in living animals and are capable of creating an OCT signal stronger than the background signal of ocular tissues.

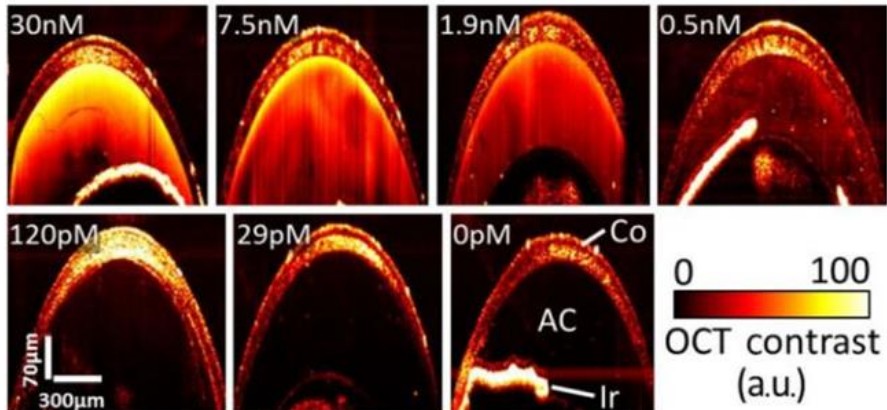

**Figure 4.** 3–5 μL of GNR-850 at concentrations from 30 nM to 0 nM were injected in the ACs of mice (N = 12). The control group was injected with Matrigel. Mice injected with 29 pM of GNR-850 showed similar contrast to control mice, whereas concentrations ≥120 pM led to a distinct detectable OCT signal [90].

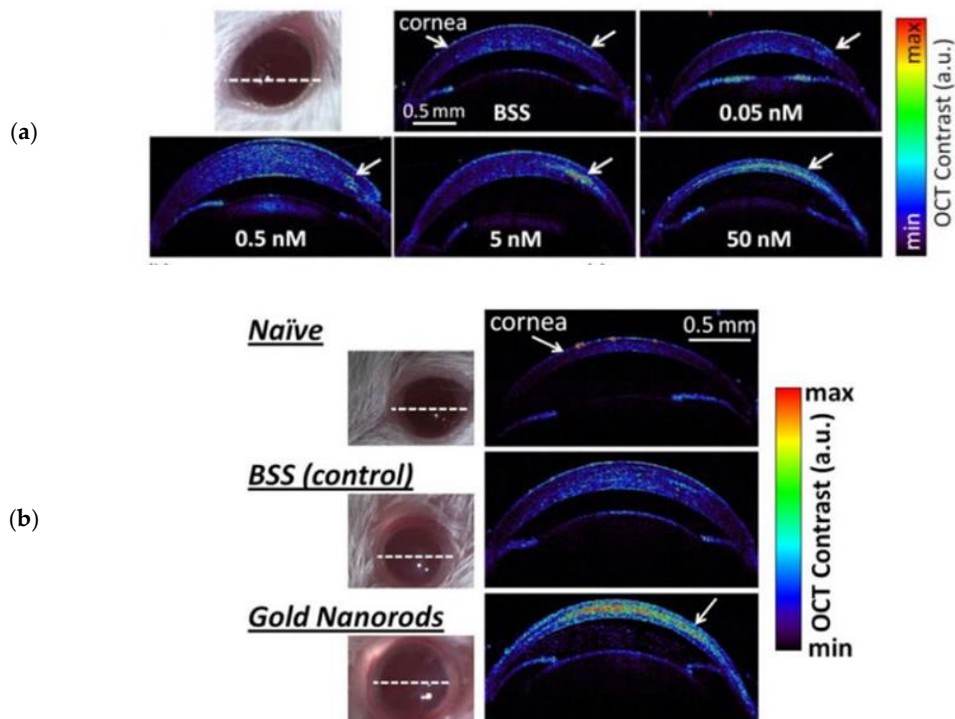

**Figure 5.** (**a**) Mice corneas injected with 10 μL GNR-780 at 50 nM (bottom) compared to control mice corneas injected with 10 μL of BSS (middle) and mice corneas not injected with anything (upper). OCT cross sectional images showed the high contrast by GNR-780 in the cornea; (**b**) mice corneas (N = 4) injected with 5 μL of GNRs at concentrations from 50 nM to 0.05 nM and control mice injected with BSS. The concentration of 0.05 nM had a contrast similar to control mice, whereas the concentration of ≥0.5 nM created a distinct detectable signal [90]. Copyright © 2014 Royal Australian and New Zealand College of Ophthalmologists. ClinExperiment Ophthalmol. Author manuscript; available in PMC 27 Jul 2016. Published in final edited form as: Clin Experiment Ophthalmol. May–Jun 2015; 43(4): 358–366. Published online 12 Feb 2015. doi:10.1111/ceo.12299.

Bioconjugated GNRs (~110 × 32 nm) were used as exogenous OCT CAs to image single cells and vessels in vitro and in vivo, in mice retinae, by Sen et al. [91]. GNRs functionalized with anti-mouse CD45 (GNR$^{CD45}$) were used to label mouse leukocytes and mPEG–GNRS (GNR$^{mPEG}$) were used to determine the in vivo sensitivity inside retinal vessels. In vitro, GNR$^{CD45}$-labeled leukocytes were imaged with OCT and had a significantly higher scattering intensity compared to the unlabeled cells. To investigate the detection sensitivity of the NPs inside the retinal vessels in vivo, GNR$^{mPEG}$ (10 nM) was injected intravenously into living mice. An increase in OCT intensities could be detected at a GNR concentration as low as 0.5 nM, and GNR$^{mPEG}$ circulated in the blood for ~4 h. These results indicate that GNRs can effectively serve as OCT contrast agents for retinal imaging and can be detected at a sensitivity of ~0.5 nM after IV injection.

Sandrian et al. [92] performed intravitreal injections of Au nanorods (aspect ratio: ~3.4) to determine their eligibility as CAs for OCT, as well as their potential ocular inflammatory effects on the eyes of mice. Images were taken before and approximately 30 min and 24 h after the injection. Unconjugated PSS–AuNRs (Au nanorods coated with poly(strenesulfate)) and Ab–AuNRs (Au nanorods coated with anti-CD90.2 antibodies) were chosen for the injections. As seen in Figures 6 and 7, an enhanced backscattered signal was noted in the vitreous of the AuNR-injected mice, when compared to baseline images and images obtained from mice that received PBS injection. A day after the GNR injection, the increased contrast could still be observed. These results indicate that AuNRs can function as CAs and be imaged within the vitreous. Nevertheless, the intravitreal

injection of bare GNRs resulted in the creation of opacities of unknown nature and were attributed to the immunological response to the GNR injection.

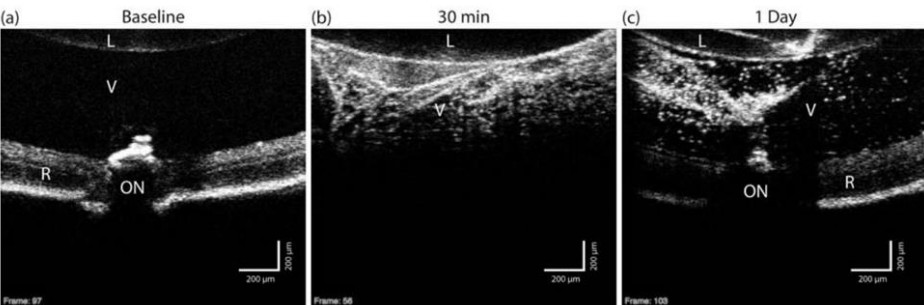

**Figure 6.** Single cross-section OCT images (**a**–**c**) of a single living mouse, before and after PSS–AuNR injection in the vitreous. Signal increase is to be noted. However, 30 min after the injection, the imaging of retina is unclear. The lens (L), the optic nerve (ON), the retina (R), and the vitreous (V) are visible in the images [92]. Copyright © 2012, BMJ Publishing Group Ltd. All rights reserved. Br J Ophthalmol. Author manuscript; available in PMC 22 Jul 2013. Published in final edited form as: Br J Ophthalmol. Dec 2012; 96(12): 1522–1529. Published online 19 Oct 2012. doi:10.1136/bjophthalmol-2012-301904.

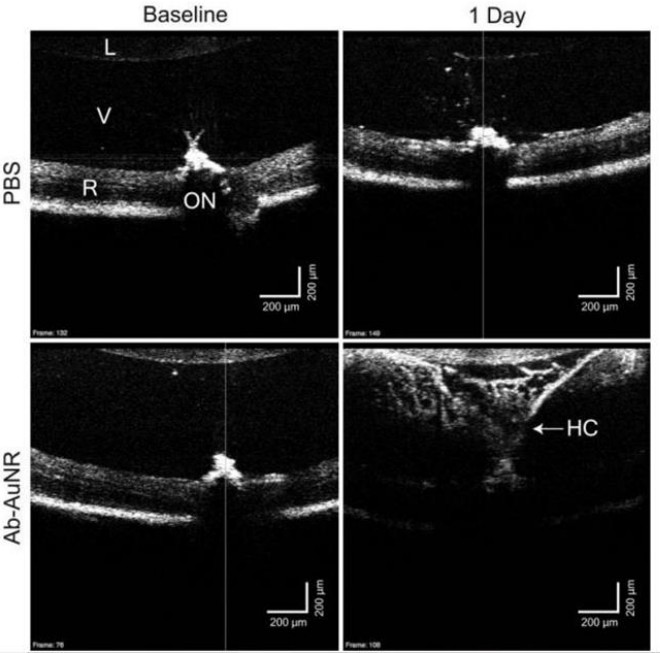

**Figure 7.** Baseline and one day post-injection OCT images from PBS–AuNR- and Ab–AuNR-injected mice. After the gold nanorod injection, significant signal enhancement can be seen in the vitreous in comparison to pre-injection and the sham injection images. The shadow occurring from the strong vitreous signal reduces the signal from the retina. The hyaloid canal (HC), the lens (L), the optic nerve (ON), the retina (R) and the vitreous (V) are visible in the above images [92].

The formation of opacities in OCT images and this "shadowing effect" was also observed by Gordon et al. [93] who performed intravitreal injections of bare Au nanorods (CTAB-coated) in mice. Within a few minutes of injection, an amorphous opacity became apparent and retinal visualization was obscured. The surface CTAB was then displaced with PEG; in this case, the injection did not cause these morphological changes. Even though PEG-coated GNRs seemed to be viable agents for in vivo imaging experiments, they were not able to reach the retina through the vitreous due to their size.

In view of the above results, Gordon et al. continued their study with intravenous injections using a laser-induced choroidal neovascularization (LCNV) model [94]. Mice

were injected with either PBS or targeted AuNRs, in order to estimate the AuNR accumulation in the lesions and the potential target effect via functionalization. Even though AuNRs reached the LCNV lesions and the functionalization was effective for targeting, they were not able to be detected using OCT due to the background noise of the tissue.

Lapierre-Landry et al. [95] proposed the use of photothermal OCT (PT-OCT) to image endogenous (melanin) and exogenous (Au nanorods) absorbers in the eyes of pigmented and albino mice. PEG-coated AuNRs or saline (control group) were IV-injected in pigmented mice with an LCNV model, and images were received 6 h later. A statistically significant ($p < 0.05$) enhancement in the PT-OCT signal was observed in the LCNV lesions of the group injected with the Au nanorods in comparison to the control group. This study showed that gold nanorods can accumulate in these lesions and be identified by an OCT-based system.

Song et al. [96] synthesized 160 nm-sized Au nanodisks to evaluate them as OCT CAs for retinal imaging in mice eyes. Nanodisks are considered to have a better scattering ability compared to AuNPs, since their signal does not depend on the direction or polarization of the light source [97]. The threshold concentration for an OCT signal following the intravitreal injections was found to be as low as 1 pM, whereas concentrations of 0.1 pM showed a minimal signal, similarly to when distilled water was administered. At higher concentrations (10 pM), the signal in the mice vitreous bodies was even stronger.

The use of AuNPs has also been proposed for the monitoring of transplanted cells using multimodal imaging [98]. Cell replacement therapy has been presented as a method to restore vision in retinal degenerative diseases [99]. Photoreceptor precursors (PRPs) are transplanted subretinally, but their tracking can be challenging. For this purpose, Chemla and co-workers suggested the use of spherical AuNPs (diameter: 20 nm) for the monitoring of PRP cells through CT, OCT, and fluorescence fundus imaging. More specifically, AuNP- and fluorescently labeled PRPs were transplanted in the vitreous and subretinal space of pigmented rats and were monitored in vivo. OCT imaging at 24 h could detect PRP clusters of cells subretinally. At days 7 and 30, the imaging of small cellular clusters that migrated from the subretinal space toward the inner layers of the retina was possible, indicating the ability of prolonged monitoring.

### 2.3.2. Gold Nanoparticles as Contrast Agents for Photoacoustic Imaging
In Vitro Studies

Gold nanostars were designed by Raghavan et al. [100] to investigate contrast enhancement in photoacoustic imaging using phantoms. These anisotropic NPs (tip-to-tip: 120–150 nm, branches length: 35–40 nm) exhibited longitudinal and transverse plasmon peaks. Using 1064 nm and 700 nm lasers, PA images of tissue phantoms with and without the addition of nanostars were compared. Signal enhancement was observed when the concentration of Au nanostars increased. Of particular importance was the fact that a potent signal was produced in the wavelength of 1064 nm, which allows deep tissue penetration in vivo.

In vitro studies were also conducted by Bayer et al. [101] who designed silica-coated Au nanorods for molecular PAI. $SiO_2$−AuNPs (thickness: 40 nm) were functionalized with monoclonal Abs, which targeted specific proteins/cell-receptors that were over-expressed by cells within the tissue phantoms. Separate cell inclusions of the tissue phantom were identified with the use of a multispectral PAI system, due to the different wavelengths of the $SiO_2$−AuNPs used to label the cell types. Therefore, silica-coated Au nanorods were proved to promote molecular imaging through a multiplex photoacoustic system.

AuNRs of small dimensions absorbing in the NIR-II were engineered by Chen et al. [102], using a seedless method. These miniature AuNRs (smallest AuNR: $8 \pm 2$ nm by $49 \pm 8$ nm) were tested as CAs for PAI and their performance was compared with that of regular-sized AuNRs ($18 \pm 4$ nm by $120 \pm 17$ nm). It was seen that miniature AuNRs not only produced a ~3.5-fold stronger photoacoustic signal but were also characterized by better photothermal stability under nanosecond irradiation. Chen and co-workers attributed this phenomenon to the greater surface-to-volume ratio of miniature NPs, which boosts heat transfer.

Ex Vivo Studies

Raveendran et al. [103] examined the function of gold nanocages (AuNcgs) as contrast agents for PAI using enucleated porcine eye models with the purpose of improving the diagnosis of ocular diseases, such as uveal melanoma, an intraocular cancer. Separate tubings filled with AuNcg (wall thickness: $5 \pm 2$ nm, average edge length: ~65 nm) solutions at different concentrations were used to obtain PA and US images and to understand the effect of the concentration on the signal. It was found that the amplitude of the photoacoustic waves was directly proportional to the concentration of the AuNcg solution and that the signals were produced by the nanocages under pulsed optical excitation. Furthermore, PA and US images before and after the injection showed that there was an increase in the PA signals from 17.6% to 81.4%. Therefore, the idea of using AuNcgs for ocular photoacoustic imaging produced encouraging results.

In Vivo Studies

Kim et al. [104] investigated the potential of AuNPs to work as both a therapeutic and diagnostic means for intraocular tumors. AuNPs coated with fucoidan (Fu) and conjugated with doxorubicin (Dox), a chemotherapeutic drug, were able to act as anti-tumor agents due to their chemo-photothermal properties in vitro and in vivo. Using rabbit models with VX2 tumors, the feasibility of Dox-Fu@AuNPs to function as CA for photoacoustic imaging was tested. Imaging before and after the intratumoral injection of 100 µL of Dox-Fu@AuNPs (200 µg/µL) was performed in vivo. The tumor that had received the injection produced stronger PA signals compared to images before the injection. Moreover, the post-injection image allowed the visualization of more than two-fold deeper tissue ($p < 0.001$), and the tumor margins were clearly viewed (Figure 8). This study proved that AuNPs can be useful as tools for both the treatment and imaging of intraocular tumors, as they increase the PA contrast and aid in determining the tumor margins.

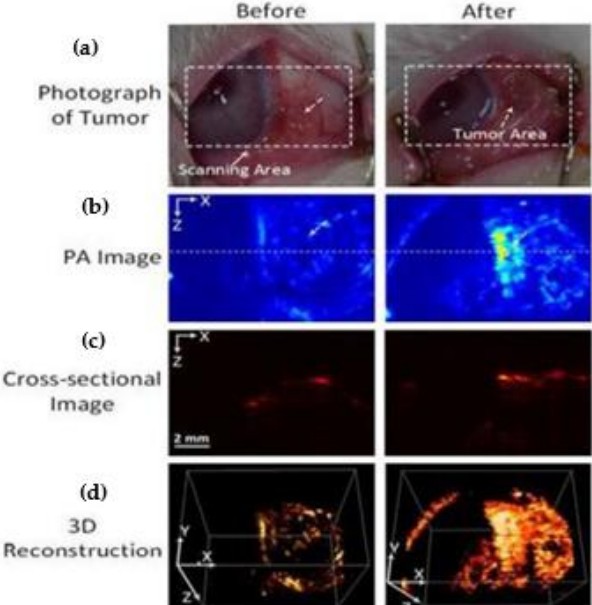

**Figure 8.** Intratumoral injections of Dox-Fu@AuNPs were performed in rabbit eyes. (**a**) On the top panel, photographs of the tumor are visible before and after the injections; (**b**) photoacoustic images (**b**) disclosed enhanced signal after the injection ($p < 0.05$); (**c**) cross-sectional images; (**d**) 3D reconstructions [104]. Kim H., Nguyen V. Phuc, Manivasagan P., Jung M. Jung, Kim S. Won, Oh J., Kang H. Wook Doxorubicin-fucoidan-gold nanoparticles composite for dual chemo-photothermal treatment on eye tumors. Oncotarget. 2017; 8: 113719–113733. Retrieved from https://www.oncotarget.com/article/23092/text/ (accessed on 7 December 2022). This is an open access article distributed under the Creative Commons Attribution License.

2.3.3. Gold Nanoparticles as Contrast Agents for a Multimodal OCT & PAI System
In Vitro Studies

Two different-sized nanodisks were synthesized in a stacked form by Wi et al. [105] and assessed as CAs for a bimodal PA and OCT system. The smaller nanodisk (80 nm) was placed on top of the larger one (180 nm) and presented LSPR peaks at two separate wavelengths. Gold nanodisks with small diameters (<100 nm) are more appropriate for light absorption and therefore can be utilized in OCT imaging, whereas the ones with larger diameters (>100 nm) are more appropriate for light-scattering and can be effectively used in PAM [106]. Thus, the stacked Au nanodisks could be applied as a bimodal contrast agent for both OCT and PAM.

For their study, Wi and co-workers used four silicon tubes that contained: blood phantom, AuNRs (PAM-CA; resonant wavelength: 650 nm), stacked Au nanodisks (bimodal CA), and Au nanospheres (OCT-CA; resonant wavelength: 850 nm). Quantitative measurements showed that only the stacked Au nanodisks were detected by both OCT and PAM, and furthermore the PAM intensity of a single stacked Au nanodisk was two-fold larger than that of an Au nanorod. Therefore, Au nanodisks in a stacked form can effectively function as sensitive contrast media and improve both PAM and OCT images.

Table 1 summarizes the data from the in vitro studies using gold nanoparticles as contrast agents for both OCT and PAI.

**Table 1.** In vitro studies examining the use of gold nanoparticles as contrast agents for OCT & PAI.

| Imaging Modality | AuNP Type | Dimensions | LSPR Peak | Tissue/Cells | Results | References |
|---|---|---|---|---|---|---|
| Time-domain OCT (890 nm) | Cylinder-like-shaped nanorods | Diameter: ~20 nm | 750 nm, 912 nm | Polyacrylamide-based phantoms | Strength of signal dependent on LSPR overlap between the AuNPs and the operating system, Both suspensions enhanced the signal contrast vs. water | Troutman et al. [68] |
| Fourier-domain OCT (~840 nm) | Nanorods coated with PEG and Tat peptide | Diameter: 10 nm | 870 nm | Intralipid tissue phantoms (mimicking retinal tissue), RPE cells | The presence of AuNPs caused a red shift in OCT images, RPE labeling with AuNRs allowed their tracking | Jia et al. [69] |
| OCT (1300 nm) | Nanorods | Length: 88 ± 5 nm, diameter: 10 ± 2 nm (aspect ratio: 8.8) | 1320 nm | Agar–TiO$_2$ phantom | AuNRs can effectively act as absorption-based CAs | Ratheesh et al. [70] |
| Dual-band OCT system with two separated bandwidths (830 nm & 1220 nm) | Nanorods | Average Length: 75 nm (large nanorods), 48 nm (small nanorods) | 868 nm (large nanorods), 835 (small nanorods) | Agar phantom | The strongest signal intensities at both bandwidths were produced with nanorods large sample, whereas were not detected with nanorods small sample | Rawashdeh et al. [71] |
| OCT (830 nm) | Nanoshells | Core radius: 100 nm, Shell thickness: 20 nm | 830 nm | 1 mm pathlength cuvette with solutions of nanoshells in water, saline & microspheres | Grayscale intensity of saline solution: 247, Grayscale intensity of nanoshells solution: 160 | Loo et al. [107] |

<div style="text-align:center">**Table 1.** *Cont.*</div>

| Imaging Modality | AuNP Type | Dimensions | LSPR Peak | Tissue/Cells | Results | References |
|---|---|---|---|---|---|---|
| Time-domain OCT (1310 nm) | PEGylated mono-layered nanoshells | Core diameters: 126–291 nm shell thicknesses: 8–25 nm | N/A | Water & turbid tissue-simulating phantoms | The strongest backscattering was produced by NPs with core diameter of 291 nm and shell thickness of 25 nm | Agrawal et al. [75] |
| OCT (900 nm) | Silica–gold nanoshells | Silica core: 150 nm, Gold shell thickness: 25 nm | 850–950 nm | Agar biotissue phantoms | Nanoshells penetrating the phantoms caused the intensification of the signal | Zagaynova et al. [76] |
| Phase-sensitive OCT (1315 nm) | Nanoshells | Core: 120 nm, shell thickness: 16 nm | 780 nm | Phantoms | High contrast between the phantoms with and without nanoshells | Adler et al. [77] |
| Spectroscopic OCT (716 nm) | Nanocages | average edge length: ~35 nm | ~716 nm | Gelatin-made tissue phantoms | The absorption cross section with nanocages presented a ~5 orders larger magnitude than conventional dyes | Cang et al. [78] |
| OCT (1325 nm) | Spherical-, cubic- & star-like-shaped NPs coated with amino acid molecules | Various | 520–110 nm | Water & agarose phantoms | Star-shaped AuNPs with less than 150 nm size produced the best contrast in water as well as in agarose phantoms. | Ponce-de-Leon et al. [79] |
| Spectral- domain OCT & Doppler OCT (930 nm) | Nanostars | Tip-to-tip: 50, 82, 100 & 120 nm | 710–830 nm | Glass capillaries | 120 nm-sized nanostars produced the best contrast enhancement | Bibikova et al. [80] |
| Multispectral photoacoustic imaging (1064 nm, 700 nm) | Nanostars | Tip-to-tip: 120–150 nm, Branches length: 35–40 nm | Transverse: 700 nm, Longitudinal: 1050–1150 nm | Tissue phantoms (mix of 2% agarose in water with 1% intralipid) | Signal enhancement was observed as the concentration of Au nanostars increased | Raghavan et al. [100] |
| Multispectral photoacoustic imaging (700–910 nm) | Silica-coated nanorods (SiO₂–AuNRs) | Thickness: 40 nm | 780 nm, 830 nm | Tissue phantoms (8% *w/v* gelatin, 1.2% *w/v* 5 μm diameter silica scatterers) | The use of AuNRs allowed the identification of separate cell inclusions of the tissue phantom | Bayer et al. [101] |
| Photoacoustic imaging | Miniature nanorods | Smallest: 8 ± 2 nm by 49 ± 8 nm | 1000–1200 nm | Tube phantoms | Miniature AuNRs produced a ~3.5-fold stronger PA signal vs. regular-sized AuNRs, better photothermal stability under nanosecond irradiation | Chen et al. [102] |
| Bimodal PA and OCT system | Stacked nanodiks | Top nanodisk: 80 nm, Bottom nanodisk: 180 nm | Top nanodisk: 630 nm, Bottom nanodisk: 850 nm | Agarose tissue phantoms | Only stacked Au nanodisks were detected by both OCT and PAM vs. nanospheres and AuNRs, PAM intensity of a single stacked Au nanodisk was two-fold larger than that of a AuNR | Wi et al. [105] |

Silica-coated nanorods (SiO₂–AuNRs) values: thickness 40 nm, LSPR 780 nm, 830 nm.

In Vivo Studies

In 2019, Nguyen et al. [108] synthesized PEGylated AuNPs and tested their capability to serve as CAs for multimodal imaging with PAM and OCT in vivo. Colloidal AuNPs ($20.0 \pm 1.5$ nm) were intravenously injected in living New Zealand and Dutch-belted pigmented rabbits, and their retinal and choroidal microvasculature was examined. Images of the retinal and choroidal vessels were obtained before (control group) and after the injection of 0.8 mL PEG–AuNPs (2 mg/mL). The rabbits injected with PEG–AuNPs led to a stronger signal compared to the control animals, and the PA signal of the vessels reached an increase of 52% over the one without injection. To examine dynamic changes, the PEG–AuNP concentration was raised to 5 mg/mL and PAM images were acquired every minute for 14 min; the increase in the signal was found to be 82%. Finally, Nguyen and co-workers tested the ability of PEG–AuNPs to function as CAs for OCT and found that the rabbit injected with AuNPs demonstrated ~45% greater OCT intensity than the control. Overall, this study demonstrated that PEG–AuNPs can function as a multimodal CA for PAM and OCT in vivo in rabbits for the examination of retinal and choroidal microvasculature.

In another study by Nguyen et al. [109], Au nanostars functionalized with RGD peptide were tested as CAs for multimodal imaging with OCT and PAM in vitro and in vivo, using living rabbits with a CNV model. The purpose of this study was to test whether the GNS-RGD (average diameter: 30 nm) can help distinguish neovascularization from the surrounding retinal vasculature, with the hypothesis that RGD peptides target $\alpha v \beta 3$ integrin, which is expressed in CNV (Figure 9).

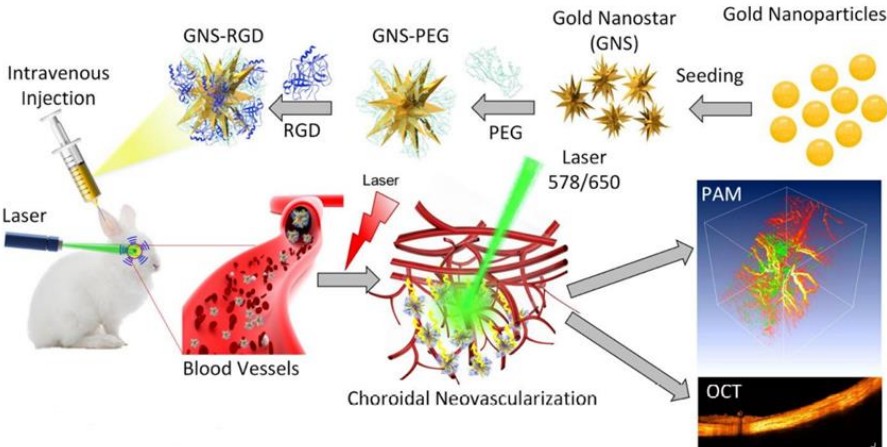

**Figure 9.** Fabricated gold nanostars functionalized with RGD peptide and examined their function as contrast agents for a multimodal OCT and PAM imaging system in living rabbits, using a laser-induced choroidal neovascularization (CNV) model [109]. Reprinted with permission from Nguyen VP, Li Y, Henry J, Zhang W, Aaberg M, Jones S, Qian T, Wang X, Paulus YM. Plasmonic Gold Nanostar-Enhanced Multimodal Photoacoustic Microscopy and Optical Coherence Tomography Molecular Imaging to Evaluate Choroidal Neovascularization. ACS Sens. 23 Oct 2020; 5(10): 3070–3081. doi:10.1021/acssensors.0c00908. Epub 30 Sep 2020. PMID: 32921042; PMCID: PMC8121042. Copyright 2020 American Chemical Society.

For the in vivo studies, rabbits were IV-injected with 400 µL of GNSs (5 mg/mL), sequential PAM images were obtained at 578 nm and 650 nm, and a high contrast between the CNV and the adjacent vasculature was created (Figure 10). The PA signal within the CNV after the nanostar injection was characterized by a 17-fold increase. GNSs were also tested as CAs for OCT and images were taken pre- and post-injection at 2 h. It was noted that the CNV was depicted with better contrast after the injection and the OCT signal reached ~167% higher intensity compared to pre-injection ($p < 0.001$) (Figure 11). In conclusion, this study showed that gold nanostars can function as high-sensitivity contrast agents for OCT and PAM and allow the CNV to be differentiated from the surrounding microvasculature.

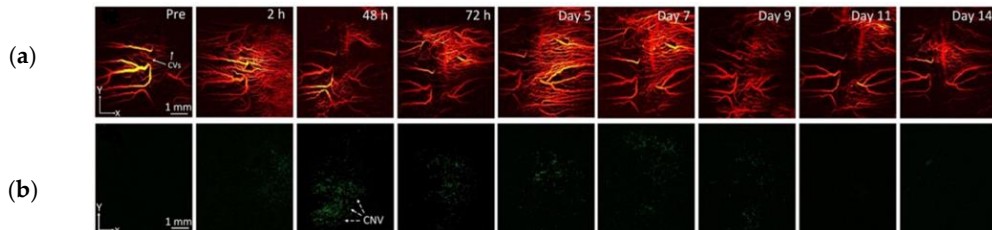

**Figure 10.** Longitudinal in vivo PAM images at different wavelengths: (**a**) 578 nm and (**b**) 650 nm before then GNS injection and 2 h, 48 h, 72 h, 5 d, 7 d, 9 d, 11 d and 14 d after the GNS injection (0.4 mL, 5 mg/mL). White arrows show choroidal vessels (CVs) and white dotted arrows show choroidal neovascularization (CNV) [109].

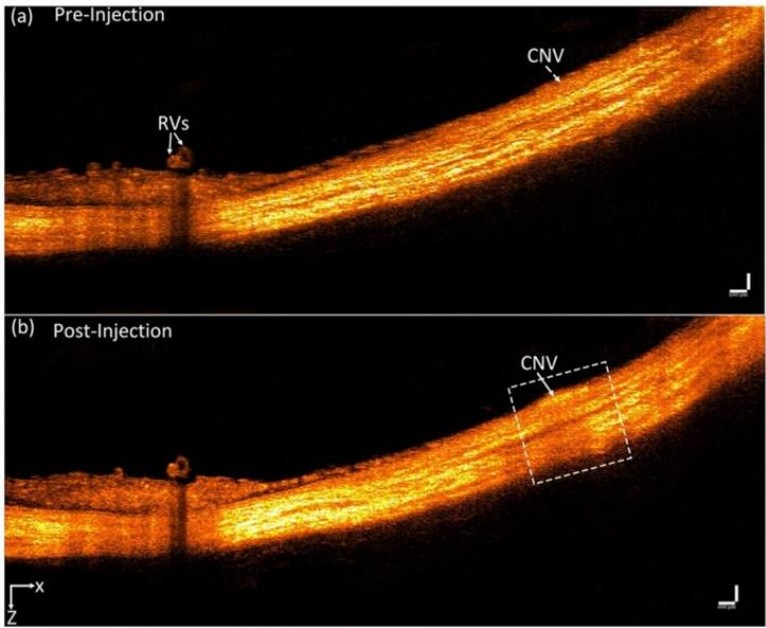

**Figure 11.** OCT image enhanced with GNS in vivo: B-scan OCT images obtained before (**a**) and 2 h after IV injection of GNS (**b**). The white arrows indicate the position of choroidal neovascularization (CNV) [109].

Recently, the same research team conducted similar experiments, but this time using chain-like gold nanoparticle (CGNP) clusters, functionalized with RGD peptides (CGNP clusters–RGD) [110]. First, 0.4 mL CGNP clusters–RGD (5 mg/mL) were administered intravenously to rabbits with laser-induced CNV, and PA images were obtained before and after the injection at 578 nm and 650 nm. The PA signal of the CNV at both wavelengths was significantly elevated compared to the images before the injection. Overall, the peak signal showed a 17-fold increase (0.11 ± 0.01 to 1.89 ± 0.1) at 24 h after the injection. OCT images were also taken before and after the IV injection of CGNP clusters–RGD at different time points. The peak OCT signal was noticed at 48 h and the contrast-to-noise ratio improved from 1 to 1.76.

Table 2 summarizes the data from the in vivo studies investigating the use of gold nanoparticles as contrast agents for both OCT and PAI.

**Table 2.** In vivo studies examining the use of gold nanoparticles as contrast agents for OCT & PAI.

| Imaging Modality | AuNP Type | Dimensions | LSPR Peak | Concentration/Dose | Administration | Subject Type | Results | References |
|---|---|---|---|---|---|---|---|---|
| SD-OCT | PEG-coated nanorods | GNR-780: length: 43 ± 4.22 nm; diam.: 12 ± 0.25 nm GNR-850: length: 49.31 ± 6.9 nm, diam.: 12.09 ± 1.63 nm | 780 nm & 850 nm | GNR-780: 50 nM/5–10 µL (corneal stroma) GNR-850: 29 pM-30 nM/3–5 µL (AC) | Anterior chamber (AC) (GNR-850) & corneal injections (GNR-780) | Wild-type C57BL/6 mice | GNR-780: injected corneas: ×3 stronger signal vs. BSS-injected and ×7.5 vs. naïve mice; concentrations > 5 nM lead to clear contrast GNR-850: Threshold concentration for significantly enhanced signal > 120 pM | De la Zerda et al. [90] |
| SD-OCT | mPEG-coated nanorods | ~110 × 32 nm | 824–830 nm (longitudinal) | 10 nM/200 µL (in steps of 25 µL) | Intravenous injections | Nu/nu mice | Images of the retinal blood vessels showed that AuNRs were perceived at a sensitivity of ~0.5 nM | Sen et al. [91] |
| SD-OCT | Nanorods coated with poly(strenesulfate)/(PSS–AuNRs) or anti-CD90.2 antibodies/Ab-AuNRs | Aspect ratio: ~3.4 | PSS–AuNRs: 850 nm Ab–AuNRs: 857 nm | 2 µL | Intravitreal injections | C57BL/6 mice | Enhanced backscattered signal in the vitreous of the mice vs. control group, even after 24 h | Sandrian et al. [92] |
| OCT | CTAB-coated nanorods, PEG-coated nanorods, Targeted nanorods (ICAM2) | Diameter: 10 nm | 808 nm | 100 µL of the AuNR solution for IV injections | Intravitreal & Intravenous Injections | Wild-type C57BL/6 mice, LCNV model | Images after Intravitreal CTAB–AuNRs inj. showed amorphous opacity, PEG–AuNPs intravitreal inj. had no shadowing effect, IV inj. of targeted NPs caused unclear results due to retinal background noise | Gordon et al. [93] |
| PT-OCT | PEG-coated nanorods | diameter: 10 nm, length: 35 nm | 750 nm | 1.66 nM/ 100 µL | Intravenous injections | Pigmented mice C57BL/6, LCNV model | Statistically significant ($p < 0.05$) increase in the PT-OCT signal in the LCNV lesions vs the control group | Lapierre-Landry et al. [95] |

**Table 2.** *Cont.*

| Imaging Modality | AuNP Type | Dimensions | LSPR Peak | Concentration/Dose | Administration | Subject Type | Results | References |
|---|---|---|---|---|---|---|---|---|
| SD-OCT | Nanodisks | 160 nm | 830 nm | 0.1–10 pM | Intravitreal injections | C57BL/6 J mice | Threshold concentration for significant OCT enhancement vs. the control group was found 1 pM, Signal increases in a dose-dependent manner | Song et al. [96] |
| OCT | PEG-coated nanospheres | Average diameter: 20 nm | 532 nm | PRPs incubated with AuNPs at a concentration of 0.2 mg/mL | AuNP- & fluorescently labeled PRPs transplanted intravitreally & subretinally | Long-Evans pigmented rats | Prolonged monitoring of the transplanted AuNP-labeled cells was possible even after 1 month | Chemla et al. [98] |
| PAI | Doxorubicin-coated nanospheresloaded with fucoidan (Dox-Fu@AuNPs) | 101.5 ± 23.2 nm | 532 nm | 200 µg/µL/ 100 µL | Intratumoral injection in the rabbit eye VX2 tumors | New Zealand white rabbits | Dox-Fu@AuNPs-injected tumors showed stronger PA signals vs. pre-injection, X 2 deeper image depth ($p < 0.001$) | Kim et al. [104] |
| Multimodal PAM & OCT | PEG-coated nanospheres | 20.0 ± 1.5 nm | 520 nm | 5 mg/mL/ 0.8 mL | Intravenous injections | New Zealand white rabbits | The OCT & PAM signal from retinal and choroidal visualization was increased by 45% and 82% respectively vs. control group | Nguyen et al. [108] |
| Multimodal PAM & OCT | Nanostars conjugated with RGD peptide | Average diameter: 30 nm | 650 nm | 5 mg/mL/ 400 µL | Intravenous injections | New Zealand white rabbits, CNV model | Photoacoustic performance raised × 17 and OCT intensities were elevated by 167% | Nguyen et al. [109] |
| Multimodal PAM & OCT | Chain-like gold nanoparticle (CGNP) clusters conjugated with RGD peptide | Average diameter: 20 nm | 650 nm | 5 mg/mL/ 400 µL | Intravenous injections | New Zealand white rabbits, CNV model | Photoacoustic performance raised × 17 and OCT intensities were elevated by 176% | Nguyen et al. [110] |

### 2.4. Ocular Distribution and Safety of Gold Nanoparticles

2.4.1. Factors Affecting Distribution and Safety of Gold Nanoparticles

Even though gold nanoparticles offer unique properties and are promising candidates for diagnostic purposes in ophthalmology, it is of paramount importance to test their biodistribution and potential bioaccumulation and toxicity, before introducing them in clinical practice. AuNPs are generally considered to be non-toxic; however, there are also data indicating the possibility of a toxic effect [111–113].

Factors that can alter the biocompatibility of gold nanoparticles need to be studied thoroughly, including the size and concentration of the NPs [114,115]. The significance of the nanoparticle size was highlighted by Jong et al. [116], who carried out in vivo experiments by intravenously injecting Au nanospheres in rats and showing a size-dependent distribution: The smaller nanospheres with a diameter of 10 nm were spotted in various organs, including the blood, liver, spleen, kidney, testis, thymus, heart, lung and brain, whereas larger NPs were only detected in the blood, the liver and the spleen. A similar study involving the intravenous delivery of gold nanospheres of various sizes (15, 50, 100 and 200 nm) in mice was conducted by Sonavane et al. [117]. Smaller NPs presented a wider distribution in the mice organs and those with sizes below 50 nm were able to cross the blood–brain barrier (BBB).

The repeated administration of AuNPs ($12.5 \pm 1.7$ nm) was assessed by Lasagna-Reeves et al. [118] in C57/BL6 mice. AuNP solutions were given through daily intraperitoneal injections (100 µL) in various doses in different groups: 40, 200, and 400 µg/kg/day for eight days. Nanoparticle blood levels were similar among all the groups, but a dose-dependent accumulation pattern was noted in the kidneys and the spleen. Nevertheless, no weight or behavioral changes were observed among the animals, and no changes in tissue morphology or hematological and histopathological examinations occurred.

Apart from the size and dose of the administrated nanoparticles, their surface chemistry and charge are also key factors for their biocompatibility profile. The presence of cetyltrimethylammonium bromide (CTAB), a cationic detergent utilized as a stabilizing agent for the preparation of AuNPs, has been found to cause cytotoxicity [119,120]. To reduce this cytotoxic effect, Niidome et al. [121] fabricated Au nanorods modified with polyethylene glycol (PEG) and removed CTAB; in vitro results using HeLa cells indicated a reduction in cytotoxicity. When PEG-modified AuNRs were injected intravenously in mice, more than 50% were detected in the blood, whereas CTAB-stabilized Au nanorods were mostly spotted in the liver.

Therefore, modification with PEG can offer stealth properties and reduce cytotoxicity and bioaccumulation. These stealth characteristics result from the fact that PEG obstructs the binding of plasma proteins on the NP surface, helping it avoid recognition by the reticuloendothelial system. Gold nanoparticles modified with PEGs of high molecular weight (>5000 Da) were found to provide better stability and cause less toxicity than those modified with PEGs of low molecular weight (<5000 Da) [122].

The nanoparticle charge can also affect their potential toxicity. Goodman et al. [123] tested cationic gold nanospheres with a diameter of 2 nm and found that they were toxic in a cell line, whereas negatively charged ones did not cause toxicity in the same cell line.

2.4.2. Studies Investigating Ocular Distribution and Safety of Gold Nanoparticles

All the parameters analyzed above are important factors to be taken under consideration in in vivo and clinical applications of gold nanoparticles. The studies that have been presented in this review have generally provided results of non-toxicity related to the use of AuNPs, unless indicated otherwise. However, limited research has been conducted regarding AuNP compatibility in ocular cells and tissues. The in vitro, ex vivo and in vitro experiments that consider the distribution and safety issues of AuNPs regarding retinal cells and ocular tissues are presented below.

In Vitro Studies

The biocompatibility of AuNPs with different morphologies and sizes was examined by Karakoçak et al. [124] using a retinal pigment epithelial cell line (ARPE-19). Gold spheres (5–100 nm), cubes (50 nm) or rods (10 × 90 nm) were selected and the lethal dose required to kill 50% of the cells (LD$_{50}$) was calculated with an MTT assay (3-[4,5 dimethyl-thiazoly-2-yl]2-5 diphenyl tetrazolium bromide).

Results indicated that small Au nanospheres and Au nanorods decreased cell viability even at low concentration, whereas larger nanospheres and nanocubes did not significantly affect viability. Moreover, the increased surface area of spherical AuNPs was associated with decreased biocompatibility, independently of the size. Interestingly, the surface area concentration that caused the death of half of the retinal cells was reported to be similar for the spheres of various dimensions (5–30 nm).

The proliferation of ARPE-19 cells was also studied by Hayashi et al. [125] in correlation with AuNP exposure. Gold nanoparticles at different concentrations (10 μM, 100 μM, and 1 mM) were added to the growth medium and no effect on the cell growth was noticed.

Ex Vivo Studies

Kim et al. [126] studied the distribution of Au nanoparticles in the retinal layers after intravenous administration in C57BL/6 mice. They demonstrated that AuNPs can penetrate the blood–retinal barrier (BRB) after IV injection in a size-dependent matter, without inducing inflammation in the vitreous, retina, or choroid of enucleated mice eyes.

When Söderstjerna et al. [127] used an ex vivo cultured mouse retina model to analyze the exposure of 20 and 80 nm Ag- and AuNPs at low concentrations, it was seen that the NPs were distributed across all retinal neuronal layers and smaller NPs were also localized within the nuclei of cells as well as other cellular compartments. An increase in the number of pyknotic cells and vacuoles was observed in the inner plexiform layers (IPL). Apart from the above morphological changes, the NPs affected glial and microglial function, as well as inducing cytotoxicity, apoptosis and oxidative stress.

Gold nanorods were intravitreally delivered to Dutch-belted rabbits by Bakri et al. [128], to evaluate whether they can cause retinal toxicity. The visualization of the inner layers of the retina did not indicate cellular atrophy or disorganization. In both the control and treatment group, the formation of vacuoles in the cells of the ganglion layer and disorganization of the outer segments of the photoreceptors was observed to the same extent, which was attributed to autolysis. However, no toxicity in the histological analysis of the retina and the optic nerve was detected, now was any ocular inflammation seen with light microscopy.

In Vivo Studies

Kim et al. [129] used Zebrafish embryos to evaluate the effect of AuNPs functionalized with positively charged N, N, N-trimethylammonium ethanethiol (TMAT) on ocular development. Zebrafish exposed to TMAT–AuNPs developed eyes that were smaller and paler in color than normal. Furthermore, genes linked to apoptosis, such as p53 and bax, were upregulated, whereas genes linked to eye development, such as pax6a, pax6b, otx2, and rx1, were downregulated. The decreased pigmentation and the repression of ocular growth in embryos indicated that TMAT–AuNPs are capable of inducing unfavorable effects on eye development in mammals.

After confirming the utility of gold nanorods as contrast agents for OCT and visualizing them in the vitreous of mice, Sandrian et al. [91] proceeded to examine the potential inflammatory effect of intravitreal GNR injections. Mice that received GNR injections were characterized by a significant increase in CD45+ cells in the eye. The increase in the number of leukocytes, predominantly neutrophils, was attributed to ocular inflammation.

Intravitreal injections were also performed by Olson et al. [130] to assess the potential changes in retinal electrical activity in brown Norway rats. ERGs before and six weeks after the injections showed no statistically significant difference in any of the steps of the ERGs.

Therefore, the injection of the AuNPs did not elicit any electrophysiological changes and did not affect the electrical function of the retina in rats.

Song et al. [95] delivered gold nanodisks intravitreally to mice and proved that GNDs could enhance imaging and inhibit neovascularization without causing any toxicity or inflammation in vivo. Then, the experiments were continued in order to evaluate the clearance of gold nanodisks in vivo, using OCT imaging. GNDs (diameter: 160 nm) were injected within the vitreous of mice at concentrations of 10 pM. Imaging 6 h post-injection showed that the gold nanodisks were scattered all over the vitreous, whereas at day seven they were only detected in the hyaloid canal. Finally, 14 days after the injection, the gold nanodisks were not to be detected or accumulated any longer.

Apart from intravitreal delivery of AuNPs, subretinal injections have also been performed to study the distribution of the NPs in the retina. More specifically, Hayashi et al. [125] adsorbed goat IgG antibodies on gold nanoparticles and injected them into the subretinal space of pigmented Dutch-belted rabbits. Fundus photographs were obtained at seven days, one month and three months after the injection. Images one-month post-injection showed mild pigmentation in the RPE and retinal degeneration, which were more noticeable at three months. However, similar results were obtained when IgG with PBS or AuNPs were injected instead of IgG-adsorbed GNPs. One week after the injection, a histological analysis showed the presence of IgGs in the RPE, the photoreceptor layer, the ONL, and the OPL, and retinal degeneration in the ONL and the photoreceptor layer. Nevertheless, when IgGs with PBS were injected, they were detected in fewer retinal layers and milder degeneration was observed. In conclusion, they proved that AuNPs can be delivered to photoreceptor cells and the RPE, but the presence of retinal degeneration needs to be further studied.

## 3. Conclusions

Gold nanoparticles are characterized by unique physical and chemical features, and tremendous growth in scientific research exploring their ability to augment bioimaging has been noted over the last decades. AuNPs exhibit interesting optical features attributed to the localized surface plasmon resonance effect (LSPR), whose parameters can be easily adjusted by tailoring the geometry and size of the nanostructures and can be shifted from the visible to the near-infrared region of the spectrum. In the case of optical coherence tomography, gold nanoparticles can enhance the backscattering of the light and improve image quality, and regarding photoacoustic imaging, they can increase the light absorption and, thus, improve photoacoustic performance. However, early diagnosis can often be challenging in clinical practice, and molecular imaging needs to be encouraged. As a result, gold nanoparticles have been investigated through in vitro, ex vivo, and in vivo studies in animals, for their eligibility as contrast adjuvants in the above-mentioned modalities. However, before progressing from bench to bedside, there are challenges to be addressed and dealt with. The in vivo studies investigating the above effects have been recently introduced and are still limited in number. Further research on ocular tissues should be encouraged in order to fully understand the capabilities of gold nanoparticles in this field and move forward to the clinical translation of their promising results. Additionally, the ocular distribution and potential toxic effects need to be studied in detail, since the current data on their biocompatibility are conflicting. As a result, these unaddressed issues should be thoroughly examined before applying gold nanoparticles in the clinical practice of ophthalmology.

**Author Contributions:** Conceptualization, E.P.E. and E.S.; writing—original draft preparation A.K., E.S., writing—review and editing, A.K., E.S., M.A.K. and E.P.E. All authors have read and agreed to the published version of the manuscript.

**Funding:** This research received no external funding.

**Institutional Review Board Statement:** Not applicable.

**Informed Consent Statement:** Not applicable.

**Data Availability Statement:** No new data were created.

**Acknowledgments:** This review was conducted in the Frame of Master Program entitled "Nanomedicine" for the academic year 2021–2022.

**Conflicts of Interest:** The authors declare no conflict of interest.

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
