# Peer review of "Gold Nanoparticles as Contrast Agents in Ophthalmic Imaging"

_optics, doi:10.3390/opt4010007_

Round 1
Reviewer 1 Report
In the manuscript of optics-2117918, the authors aimed to review the potential applications of gold nanoparticles for diagnostic purposes in Ophthalmology, and specifical attention had been drawn to the utilization of AuNPs as contrast agents in OCT and PAI. However, the precent status of manuscript is not satisfactory and the following comments are provided.
1. A key review article is missing.
Gold nanoparticles in ophthalmology, Med Res Rev 2019;39:302–327. by Florence Masse, Mathieu Ouellette, Guillaume Lamoureux, Elodie Boisselier (https://doi.org/10.1002/med.21509)
The auhtors should cite and evaluate the above review to clarify the contribution of your paper.
2. A section of preparation methods of different gold nanoparticles should be summerized and discussed in brief.
3. The discussion of toxicity and bio-compatibility of gold nanoparticles should be added and reorganized according to the the latest literatures.
Author Response
Dear Reviewer,
all the corrections have been performed according to yours suggestions. The changes have been highlighted with yellow colour in the text. Please find attached the statement of the correction.

Reviewer 2 Report
Authors have provided a review of application of gold nanoparticles (AuNRs) in OCT and PAI imaging with emphasizing the contrast enhancement ability of AUNRs in ophthalmic imaging. The manuscript is well organized and written generally well. However, it requires a revision based on the following suggestions.
1.Since this is a review article, authors need to include maximum number of related articles linked to the topic. Occular imaging is performed in three OCT wavelength windows: 850 nm, 1050 nm, and 1310 nm. Authors didn't mention the AuNRs with long aspect ratio for higher wavelength imaging 1310 nm as reported [1] .(https://doi.org/10.1088/2057-1976/2/5/055005) Authors are advised to include the same.
2. Authors are advised to provide a short description of OCT technology and PAI in the introduction section or as a separate section, as desired [2-3]
(https://doi.org/10.1088/1361-6560/abd669; https://doi.org/10.1117/1.JBO.24.6.066011)
3. It would be better authors also provide a brief description of localized surface plasmon resonance (LSPR). Authors are also advised to include simulation as shown in the article that{1} clearly demonstrates the shift in the LSPR peak wavelength with aspect ratio.
Author Response

(The authors gave the same response as above.)

Round 2
Reviewer 1 Report
The authors have addressed the comments from the reviewers, so I recommend it to be accepted for publication in Optics.